# Ion Selective Amperometric Biosensors for Environmental Analysis of Nitrate, Nitrite and Sulfate

**DOI:** 10.3390/s20154326

**Published:** 2020-08-03

**Authors:** Niels Peter Revsbech, Michael Nielsen, Deby Fapyane

**Affiliations:** 1Aarhus University Centre for Water Technology, Department of Biology, Aarhus University, Ny Munkegade 114-116, 8000 Aarhus C, Denmark; dfapyane@bios.au.dk; 2Department of Sensor Productions, Unisense A/S, Tueager 1, 8200 Aarhus N, Denmark; min@unisense.com

**Keywords:** bacterial biosensor, enzyme biosensor, long-term stability, environmental monitoring, nitrate, nitrite, sulfate

## Abstract

Inorganic ions that can be redox-transformed by living cells can be sensed by biosensors, where the redox transformation gives rise to a current in a measuring circuit. Such biosensors may be based on enzymes, or they may be based on application of whole cells. In this review focus will be on biosensors for the environmentally important ions NO_3_^−^, NO_2_^−^, and SO_4_^2−^, and for comparison alternative sensor-based detection will also be mentioned. The developed biosensors are generally characterized by a high degree of specificity, but unfortunately also by relatively short lifetimes. There are several investigations where biosensor measurement of NO_3_^−^ and NO_2_^−^ have given new insight into the functioning of nitrogen transformations in man-made and natural environments such as sediments and biofilms, but the biosensors have not become routine tools. Future modifications resulting in better long-term stability may enable such general use.

## 1. Introduction

Concentrations of many inorganic ions in the environment can be determined by electrochemical techniques. Various voltammetry techniques are thus widely used to measure ions such as heavy metals that can be redox-transformed at moderate potentials on an electrode [1]. Voltammetry is rarely used for longer term monitoring, but in situ monitoring of dissolved sulfide over timespans of several days by use of a gold-amalgam electrode has been demonstrated [2]. Novel developments in instrumentation and electrode technology may, however, improve the possibilities for continuous on-line analysis [3]. Potentiometric ion-selective electrodes based on ionophores [4,5] are often chosen for monitoring of ions in aqueous environments. Very selective ionophores are available for several inorganic ions, such as H^+^, K^+^ and Ca^2+^ [4] and CO_3_^2−^ [6], and it is thus possible to apply electrodes for these species even in seawater. For analysis of low-salt media such as freshwater it is also possible to measure ions like NH_4_^+^, NO_2_^−^, and NO_3_^−^ by use of ionophore-based ion-selective electrodes [4], but concentrations of interfering species are often too high to allow for measurement of environmentally relevant low concentrations [5]. Advances in potentiometric ion selective electrode technology have been reviewed recently [5,7].

One option to create very selective sensors for ions is to apply enzymes or whole cells, i.e., to develop biosensors. Gene expression may give off some type of quantifiable signal when cells are exposed to some environmental factor, such as mercury ions [8]. The signal analyzed is often in the form of light emission or production of fluorescent protein (GFP) by genetically modified organisms, and the signal producing cells may be housed in a physical device, i.e., a biosensor, with optical signal transduction [9]. A main problem by use of gene expression for analyte detection is a response time of at least 30 min, and usually hours, at optimum temperature, and even longer for signal decay to base level. The slow response makes biosensors based on gene expression unsuitable for many applications. In this paper the focus is on biosensors where the analyte is either oxidized or reduced by a biological component, giving rise to a current in a measuring circuit. The environmentally most relevant inorganic ions that can be biologically redox transformed are NO_3_^−^, NO_2_^−^, NH_4_^+^, HCO_3_^−^, HS^−^ and SO_4_^2−^. Very good purely electrochemical sensors are available for the carbonate system (CO_2_ [10,11] and CO_3_^2−^ [6]) and for H_2_S [12]. No practically applicable NH_4_^+^ biosensor has been described, so focus will here be on NO_3_^−^, NO_2_^−^ and SO_4_^2−^.

## 2. Enzymatic Biosensors for Nitrate and Nitrite

Nitrate can be directly reduced to nitrite at various cathode materials [13,14,15,16,17,18], but despite the thermodynamic feasibility of the reduction it has proven difficult to obtain electrodes with sensitive and reproducible signals. The surface of a bare electrode is easily poisoned by medium constituents which is an obvious disadvantage for long term stability of the signal [19,20], and oxygen interferes strongly.

To facilitate the transfer of electrons between cathode and nitrate and to minimize reduction of other species, several attempts have been made to utilize the catalytic role of nitrate reductase (NR) in conjunction with an electron carrier—a mediator. Nitrate reductases from bacterial [19,21] as well as eukaryotic sources [19] have been utilized. In living organisms NADH supplies the electrons used by NR for nitrate reduction but use of NADH in nitrate biosensors is impractical due to instability of NADH, also resulting in fouling of the electrode surface [22]. Kirstein et al. used membrane-bound NR receiving electrons from 19 different types of mediator molecules, including phenothiazine, triphenylmethane, sulfophthalein and viologen groups adsorbed onto a graphite electrode. They found that different mediators gave different apparent *K_M_*, values for the enzyme [21]. Amperometric biosensors using NR have dynamic linear ranges extending up to 500 µM with LOD of 0.5–5 µM, depending on the immobilization methods and mediator used [21]. A general problem is that the needed negative potential also reduces O_2_, implicating that analysis should be performed under anoxic condition [21,23,24,25,26]. Anoxic conditions can, however, quite easily be obtained by addition of O_2_ scavengers [27] to water samples, even under field conditions. As the enzyme activity and kinetics are dependent on pH, the pH should be kept constant near optimum pH for the specific NR. Due to enzyme degradation or leaching, NR based biosensors often have short shelf lifetimes varying from 1–2 [21] to 7 days by dry storage [24], but appropriate immobilization in polymers such as pyrrol-pyridium and PVA may extend shelf lifetime to 3 weeks [25] and even 1 month [28]. It was not possible to find reports of extensive use of NR based biosensors for environmental monitoring, indicating that their measuring characteristics fall short of potentiometric ionophore-based electrodes (that have widespread applications in waste water treatment [29]), UV-absorption based optical sensors from for example Sea-Bird Scientific (that can be used in transparent freshwater and seawater environments [30]) and bacteria-based biosensors (see below).

Electrochemical determination of NO_2_^−^ without the help of biological catalysis can be performed by application of various electrode materials and mediators, and linear calibration curves over large concentration intervals can be obtained [31]. A general problem with detection by reduction is, however, that O_2_ and NO_3_^−^ interfere, invalidating the approach for environmental monitoring. A more promising approach is detection by nitrite oxidation, and this approach has been investigated by several researchers. The apparently best results were obtained by Zhou et al. [23] who developed a pyridinium-based ionic liquid (IL)-CNT composite electrode for nitrite detection by oxidation that had a low detection limit (0.1 µM) and a shelf lifetime of 100 days. No data were presented for long term stability in complex media, and even in pure buffer with 0.5 mM NO_2_^−^ the signal decreased by 6% within 1 h. It was, however, possible to regenerate the electrode by mechanical polishing.

The cathodic reduction of nitrite can be catalyzed by organic molecules as is the case for nitrate. Not only nitrite reductase (NiR) [31,32,33], but also cytochrome c-NiR [34,35], horseradish peroxidase [36], and hemoglobin [37] have been used to decrease the large overpotential for nitrite electroreduction. The electrooxidation of nitrite can be mediated by hemoglobin [38] and cytochrome c [39].

A good nitrite biosensor performance was achieved by Hong et al., using hemoglobin and Au-nanoparticles (AuNPs) as electron transfer relays for nitrite reduction. They reported a LOD of 65 nM and a sensor shelf lifetime of 1 month [40]. The applied potential for the hemoglobin or NiR mediated nitrite reduction was still −0.6 to −0.9 V (vs Ag/AgCl) at which potential O_2_ is also reduced, and analysis was therefore performed under completely anoxic condition [35,36,37,40]. The addition of organic mediators such as phenosafranin, safranin [34], phenaxinium methylsulfate [32], methyl viologen [33], or M-NMP+ and gallocyanine [34] resulted in much less reduction potential, but the necessity of oxygen removal still persisted. An electrooxidation of nitrite using hemoglobin/AuNPs/graphene seems to be a more promising procedure as such sensors were reported to be very sensitive, and with no O_2_ interference [38]. Interference was, however, found for I^−^, Br^−^, S_2_O_3_^2−^, and SO_3_^2−^ [38], preventing direct use in seawater with Br^−^ concentrations approaching 1 mM.

The major factors preventing general use of enzyme-based nitrate or nitrite biosensors are probably short lifetimes and poor stability by continuous use. Maximum shelf lifetimes at 4 °C were thus reported to be 30 days [38]. As the polarized electrodes are in direct electrical connection with the analyzed sample, medium constituents may foul the electrode surface by long term exposure, and no records of continuous use for environmental monitoring were found. For long term monitoring the polarized electrode should preferably be shielded behind a membrane that excludes most interfering and fouling species, and the most efficient shielding membranes are those that only allow small uncharged molecules to pass. The ions should thus be converted into gaseous products that can pass such a membrane, and such conversions are possible by use of microorganisms that reduce nitrate, nitrite and sulfate. In the following sensors based on this principle are described.

## 3. Bacteria-Based Biosensors for NO_2_^−^ and NO_3_^−^

Bacteria-based biosensors for nitrate (actually nitrate + nitrite, NO_x_^−^; [41]) and nitrite [42] rely on the bacterial reduction of nitrate and nitrite to N_2_O and subsequent reduction of this N_2_O in a Clark-type N_2_O sensor (Figure 1). The bacterial biomass is grown on agar plates, collected and injected directly into the sensor tip. The sensors have a built-in reservoir of bacterial growth medium that should contain a heavily fermentable electron donor such as glycerol. As compared to the influx of nitrate/nitrite into the sensor, the nutrient reservoir is huge and should sustain bacterial respiration for months. The length of the “reaction chamber” shown in Figure 1 should be sufficient to allow for first complete removal of O_2_ in the distant layers, and subsequently for complete reduction of NO_3_^−^ and/or NO_2_^−^ entering through the tip membrane to N_2_O. Short (e.g., 100 µm) reaction chambers will thus have a smaller dynamic range than long (e.g., 200 µm) reaction chambers, but the response time increases with longer chambers.

The response time increases and the dynamic range decreases when the temperature decreases. The 90% response time for a 200 µm reaction chamber is about 30 s, and the dynamic range (without applying a charge across the membrane, see later) is about 0–500 µM at room temperature [41]. There are no described interferences except for H_2_S that reversibly inactivates the sensor, and N_2_O. The sensors produce linear calibration curves and may have detection limits (LOD) of 0.1 µM. As will be presented later the linear range of NO_x_^−^ and NO_2_^−^ sensors can be tailored to just about any range by applying a charge across the front membrane. In contrast to ion-selective sensors based on ionophores, the bacteria-based biosensors also function well at salinities as high as seawater. For NO_x_^−^ various denitrifying bacteria devoid of N_2_O reductase may be applied, and by use of cold resistant (psychrotrophic) bacteria [43] such sensors may be used at low temperatures. Microsensors based on psychrotrophic bacteria were thus used to measure NO_x_^−^ profiles at high resolution in terms of both concentration and depth in sediment at 2000 m water depth off the coast of Tokyo, where the water temperature was only 2.5 °C [43]. The psychrotrophic strain actually functions very well at temperatures up to 30 °C, and this strain has also been applied in many published studies from warmer aquatic environments.

Unfortunately, psychrotrophic bacteria devoid of both nitrate and N_2_O reductase that could be applied in a nitrite biosensor have not been identified. It would be easy to create a nitrate reductase deficient mutant of the psychrotrophic denitrifyer just mentioned, but laws about genetically modified organisms would complicate practical use of sensors based on such bacteria. The only known rich sources of nitrate reductase deficient nitrite reducing bacteria are air filters rinsing airstreams rich in ammonia, such as ventilation air from piggeries. Nitrite may accumulate to concentrations above 100 mM in these filters [44], but due to the high air load they have temperatures identical to the air source, i.e., about 20 °C, which does not support growth of cold-tolerant bacteria.

Macroscale sensors have been constructed with flat end (Figure 2) penetrated by a drilled 0.3–0.5 mm diameter hole housing the bacteria. The front end of the bacterial chamber was covered by an ion-permeable membrane. The data obtained by the sensors can be of extraordinary quality, but N_2_O interference may have to be taken into account by analysis of some environments such as wastewater treatment plants, where considerable N_2_O concentrations may be found. It may thus be necessary to perform simultaneous measurements with a N_2_O sensor [45], so that the interference can be compensated for. The nitrate and nitrite biosensors have both been used for continuous monitoring of up to 2 weeks in aquatic environments (Figure 3) such as wastewater treatment plants [46,47] and lake water [48] (macroscale sensors) and for detailed microsensor profiling (Figure 4) at sub-mm scale of stratified microbial environments such as biofilms and sediments [49].

There are, unfortunately, several factors that contribute to a limited lifetime of nitrate and nitrite biosensors. Macroscale sensors thus tend to function for only 1–2 weeks of continuous monitoring, and microscale sensors only for a few days. Unisense A/S marketed NO_x_^−^ biosensors for two decades, but sales were too low for a profitable production and sales were stopped in 2019. For both types of sensors contamination with N_2_O reducing bacteria can be a problem. Especially for the microscale sensors it has not been possible to develop procedures whereby a pure culture of bacteria can be maintained within the sensor for any length of time. The multi-step assembly of the microsensors basically precludes a sterile sensor interior before inoculation with the wanted bacterium, and although procedures for addition of functional ion-permeable membranes have been developed [43], there is no guarantee that the membranes remain bacteria-tight for extended periods of time. For nitrite biosensors the problem with bacterial contaminations is even worse than for nitrate, as bacteria having a nitrate reductase seem to be omnipresent, whereas only relatively few bacteria may have a N_2_O reductase. Addition of 10 mM WoO_4_^2−^ to the bacterial growth medium inside the sensor inhibited growth of NO_3_^−^ reducing bacteria [42], but the sensors were still subject to growth of N_2_O reducing bacteria. For macroscale sensors sterilization of the sensor interior by treatment with propylene oxide was attempted [42], but after some time unwanted bacteria still invaded the sensor interior. In addition to the problems with bacterial contamination, disappearance of bacteria from the tip region of the sensor has also been observed. In combination the two problems described above have limited the lifetime of macroscale biosensors to a maximum of 2 weeks of continuous monitoring, and microscale biosensors to a few days. For the macroscale NO_x_^−^ sensors of Unisense, the chamber with bacteria could be changed in a few minutes. New chambers with bacteria could be stored for > 1 month at 4 °C, but such frequent maintenance prevented their extensive use in for example wastewater treatment.

The macroscale biosensors are able to supply an unmatched data quality by continuous NO_x_^−^ or NO_2_^−^ monitoring in natural waters and wastewater, but for practical widespread use it is necessary to find solutions to the two main limiting factors mentioned above: contamination and disappearance of bacteria in the tip region. The macroscale nitrate and nitrite biosensors constructed until now have been based on double-adhesive tape (Figure 2) for attaching the ion-permeable membrane (e.g., Nuclepore polycarbonate membrane). A better way of attaching the membrane to the body of the sensor should be developed to ensure a tight seal that prevents bacterial contamination from the outside. The active migration of bacteria away from the tip region may be prevented by using immobile bacteria. The ideal situation would be to find immobile bacteria that produce their own sticky extracellular polymer for physical entrapment. Attempts to create long-term stable sensors by entrapment in agar or alginate have until now failed (N.P. Revsbech, unpublished data). When the contamination and immobilization problems are solved the NO_x_^−^ and NO_2_^−^ biosensors should be able to function continuously for several months, until the internal N_2_O sensor stops functioning.

Fouling of sensors by organisms growing on the outside of the sensors is a problem for all environmental monitoring based on sensors. This problem is worse for the nitrite and nitrate biosensors described here than for most other sensors, as the biosensors continuously leak organic compounds from the internal bacterial growth medium to the outside. There is thus an absolute requirement for frequent rinsing of the front membrane—at least once per day in most environments. By immersion in wastewater treatment plants, the turbulence is often sufficient to keep the membrane clean so that mechanical cleaning can be avoided. For long term monitoring in other environments it may be necessary to apply automatic cleaning devices such as seen by some commercial sensors (e.g., the Seabird UV-based nitrate sensor). Also, during storage there is a need for frequent (e.g., every week by storage at 4 °C) front membrane cleaning as bacterial growth on the outside of the sensor will result in a cut-off of the essential O_2_ and NO_3_^−^ or NO_2_^−^ supply that keeps the internal bacterial culture alive.

## 4. Scaling of Sensor Response by ESC

A common feature for the sensor design shown in Figure 1 is that a charged molecule is transformed into a gas that can be measured by a Clark type sensor equipped with gas permeable membrane. That means that the biological conversion and the gas detection occur in two electrically insulated compartments. This allows for control of the ion permeability of the front membrane by applying a charge between bacterial chamber and the exterior [50,51]. At high positive tip charge, the entry through the membrane is controlled by migration and not by diffusion, and a very high sensitivity can be achieved. A schematic representation of the technique (ESC, Electrophoretic Sensitivity Control) is shown in Figure 5.

A potentiostat is used for applying the charge in Figure 5, as it can be used to assure a constant current in the ESC circuit and thus a constant ion transport through the membrane by migration. Commercial Ag/AgCl reference electrodes (e.g., Radiometer, Brønshøj, Denmark) may be used as counter electrodes by ESC on microsensors, and it was shown [51] that the current in the ERC circuit was constant over long time periods by application of a constant charge of −0.2 to +0.9 V to an internal Ag/AgCl electrode inserted into the nutrient reservoir. By use of the ESC methodology to macroscale sensors it is necessary to use large capacity counter electrodes, such as a large coil of chlorinated silver wire, and in this case a potentiostat will ensure that the current and thus the sensor sensitivity can be kept constant. An example of macrosensor (0.5 mm tip hole) calibration curves obtained by the ESC technique are shown in Figure 6. In this case a poorly permeable 0.015 µm pore size Nuclepore membrane was used, resulting in a very poor sensitivity at 0.0 V. However, when the sensor interior was charged positively at 0.9 V the sensor gave a signal of 23 nA for 3.7 mM of nitrite. When a charge of −0.2 V was applied the rise in signal was only 0.4 nA for 40 mM of nitrite, and thus a more than 600 fold lower sensitivity than for +0.9 V. The sensitivity by use of the 0.015 µm membrane, even at +0.9 V ESC charge, was actually quite low, but can be improved by use of for example a 0.05 µm membrane where the overall pore volume is much larger.

It should be noticed, however, that the calibrations were performed at 10‰ salinity (NaCl), and that variations in the imposed ERC potential results in much larger changes in sensitivity at lower salinity such as found in for example limnic waters or wastewater [51]. A 0.015 µm membrane may thus be ideal for low salinity environments. By use of the ERC technique it is possible to use the same sensor over a wide range of concentrations, applying a high positive tip charge at low concentrations, and no or even negative charge at high concentrations. As seen in Figure 6 the sensitivity at −0.2 V negative charge may be so low that measurements at −0.2 V may serve as control of baseline current.

## 5. Biosensor for Sulfate

In addition to nitrite and nitrate, bacteria may also respire with SO_4_^2−^, reducing SO_4_^2−^ to H_2_S. Clark-type H_2_S sensors have been described [12], and it should thus be possible to apply sulfate reducing bacteria for measurement of sulfate analogous to the use of nitrate reducing bacteria. Sulfate reducing bacteria are, however, anaerobes and can only to a very limited extent reduce O_2_. For construction of a sulfate biosensor it is thus necessary to apply a mixed culture where aerobes or facultative aerobes remove the O_2_. In a first attempt a sensor constructed as shown in Figure 1 was applied, but inoculation was with sewage biofilm constituting the mixed culture. The sensor had a tip diameter of 70 µm and a distance between tip membrane and H_2_S sensor of 400 µm. The H_2_S sensor was not the one previously described [12], but a similar sensor with better long terms stability obtained from Unisense A/S. The growth medium was phosphate buffered at pH 7.2 and contained 4 g sodium lactate per liter. A little of the biofilm was added to the growth medium, and after 4 days the sensor responded to sulfate (Figure 7). The zero current for the sensor was quite high, but the sensor responded quite linearly to changes in sulfate concentration. Use of controlled mix of pure cultures and possibly immobilization of bacteria in the sensor tip may result in considerably better sensor response. We hypothesize that the high zero current was due to H_2_ formed by fermentation of lactate by the mixed microbial community, and the applied Unisense H_2_S sensor is H_2_ sensitive.

Potentiometric ion-selective sensors have been described for sulfate [7], and for many purposes such sensors may be ideal. However, due to interference from Cl^−^ they cannot be used in saline waters. In contrast to the NO_x_^−^ and NO_2_^−^ biosensors described above, there should, however, be no problem by bacterial contamination of a sulfate biosensor that works with a mixed bacterial community. The use of a sulfate biosensor will be limited to oxidized environments, as external sulfide will interfere, but future improved versions might enable long-term monitoring in aerobic aquatic environments. Microscale sulfate biosensors may also be applied in freshwater sediments to obtain new knowledge about sulfate cycling. Interfering free sulfide is often not present in freshwater sediments as it is bound to iron minerals.

## 6. General Characteristics of NO_3_^−^, NO_2_^−^, and SO_4_^2−^ Sensors

The published characteristics of various sensor types are summarized in Table 1. It is, however, a major problem that the test conditions for the various sensors are very different. Detection limits (LOD) are thus as a standard given for analysis in media without interfering agents—except for potentiometric and bacteria-based sensors, where detection limits in natural media are well described. The response time if often not mentioned in the publications, and the times given are best estimates based on presented data. Effect of prolonged exposure of sensors to natural media has only been investigated for potentiometric and bacteria-based sensors, and it is thus difficult to define operational lifetimes for other sensor types. The mentioned operational lifetime by continuous operation of less than a few hours is thus best estimate. When multiple versions of a specific sensor type have been described, the mentioned characteristics are the best (fastest, most sensitive, longest lifetime) obtained. Most sensor types are affected by H_2_S, but few publications mention this interferent, and it is therefore not included in the table except for the sulfate biosensor.

## 7. Conclusions

Ionophore-based potentiometric ion-selective electrodes or optical sensing are generally preferred over alternative electrode- or sensor-based methods for NO_3_^−^ and NO_2_^−^ measurement in environmental waters. Problems with interferences preclude the use of NO_3_^−^ and NO_2_^−^ potentiometric ion-selective electrodes in saline waters, and the use of biosensors may here be an attractive alternative. No records could, however, be found of long-term monitoring or detailed environmental investigations by use of enzyme-based sensors. Bacteria-based sensors for NO_x_^−^ (NO_3_^−^ + NO_2_^−^) and NO_2_^−^ have been extensively used for environmental analysis and have given new insights into nitrogen transformations in nature, but limited lifetime has precluded extensive use in for example wastewater treatment. It should be possible to extend the lifetime of such sensors to several months by using immotile bacteria which produces sticky extracellular polymer for more robust physical entrapment. The biosensor monitoring of many aquatic environments may then become an attractive alternative to use of potentiometric ion-selective electrodes. A new biosensor for environmental analysis of SO_4_^2^^−^ is demonstrated, and it may be used to obtain new knowledge about sulfur cycling.

## Figures and Tables

**Figure 1 sensors-20-04326-f001:**
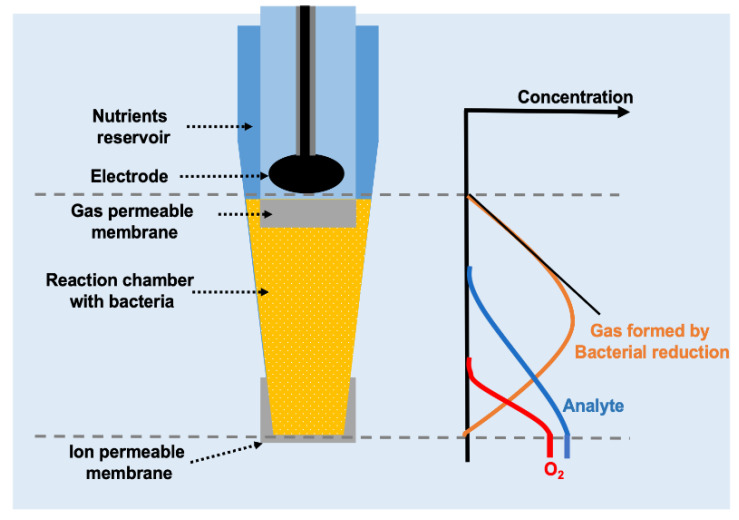
Schematic representation of biosensor based on bacteria respiring and thus reducing an electron acceptor such as NO_2_^−^ to a gas (in this case N_2_O) that can be detected by an amperometric Clark-type gas sensor. Such sensors have been realized fore NO_2_^−^, NO_x_^−^ (i.e., NO_3_^−^ + NO_2_^−^) and for SO_4_^2−^ that is reduced to H_2_S. (Original image from authors).

**Figure 2 sensors-20-04326-f002:**
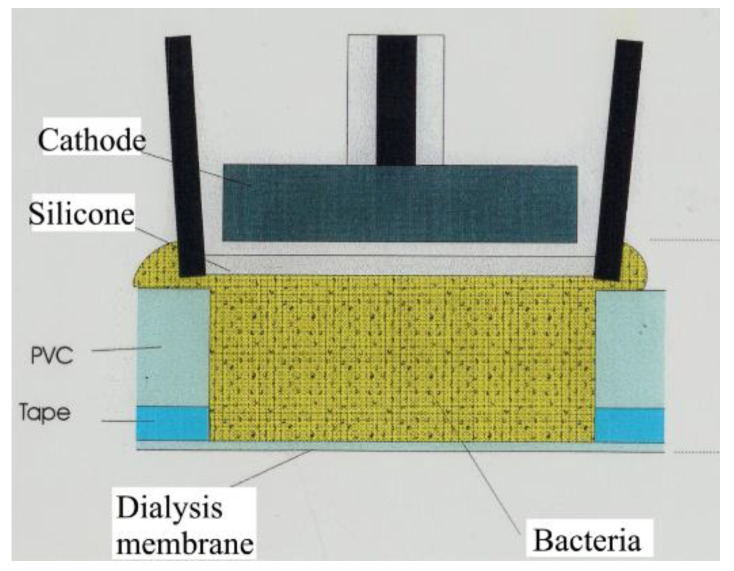
Design of a macroscale biosensor where the bacteria are housed in a 0.3–0.5 mm diameter hole drilled in a PVC or PET plate. (Original image from authors).

**Figure 3 sensors-20-04326-f003:**
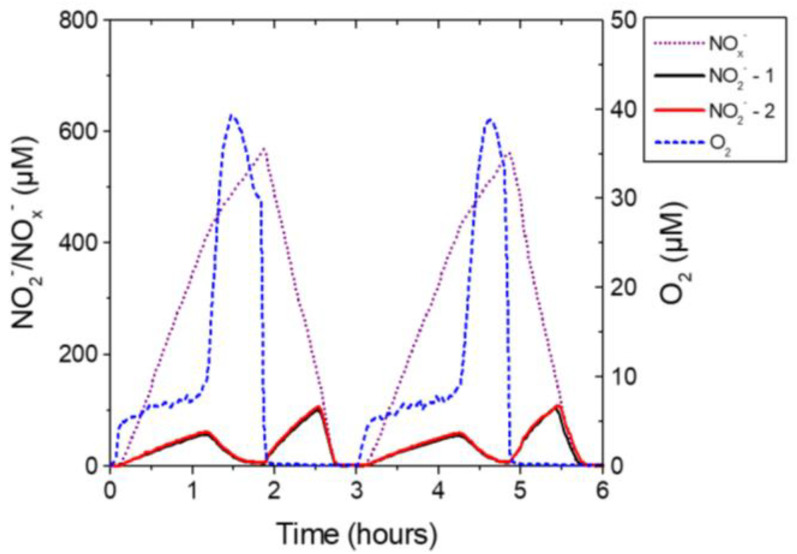
Cycles of O_2_, NO_x_^−^ and NO_2_^−^ in a pilot-scale wastewater treatment plant exposed to aeration cycles. NO_x_^−^ rose upon start of aeration and decreased when aeration was stopped. Nitrite exhibited a more complex dependence of aeration/anoxia reflecting balance between NO_2_^−^ producing and comsuming processes. Shown are traces from two NO_2_^−^ sensors showing almost identical results. (Data was obtained from the laboratory of the presented authors).

**Figure 4 sensors-20-04326-f004:**
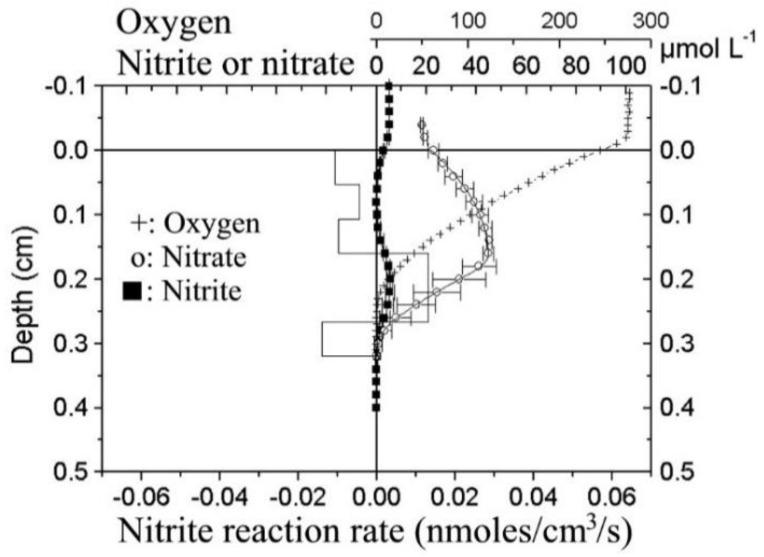
Microprofiles of O_2_, NO_x_^−^, and NO_2_^−^ in an estuarine sediment (Randers Fjord). The NO_x_^−^ and NO_2_^−^ profiles shown are averages of each 3 profiles with standard deviation indicated. Also shown are rates of NO_2_^−^ transformation obtained by diffusion-reaction modelling of the NO_2_^−^ profile. (Data was obtained from the laboratory of the presented authors).

**Figure 5 sensors-20-04326-f005:**
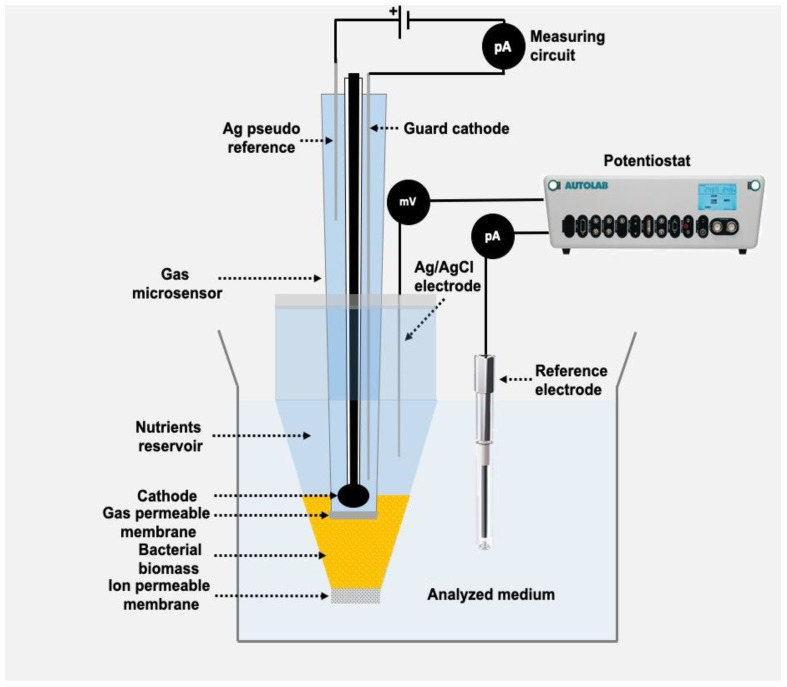
Schematic representation of the ESC technique where a potentiostat is used to assure a constant current between the Ag/AgCl electrode in the nutrient reservoir of the biosensor and an external reference. (Original image from authors).

**Figure 6 sensors-20-04326-f006:**
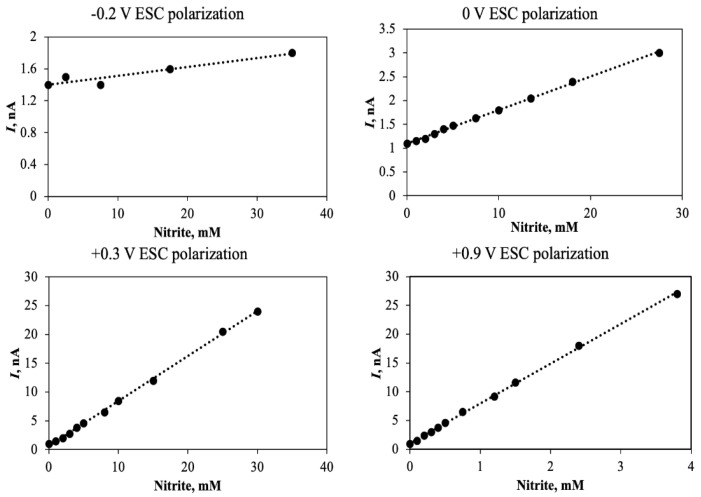
Calibration curves of a NO_2_^−^ biosensor polarized at various ESC potentials. A potentiostat as shown in Figure 5 was not applied for this experiment, but the calibration curves were still linear, indicating that the current in the ESC circuit was constant during the recording of each calibration curve that required about 1 h. (data obtained from the laboratory of the authors).

**Figure 7 sensors-20-04326-f007:**
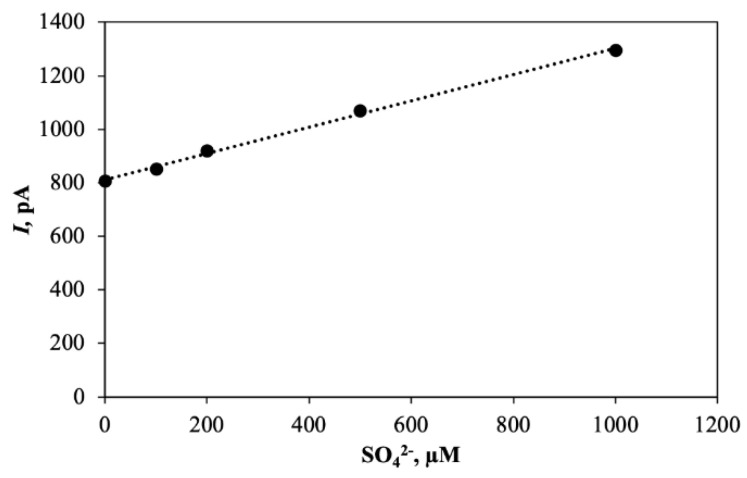
Calibration curve for SO_4_^2−^ biosensor with 70 um tip. The 90% response time was 5 min due to a long diffusion path through a 400 µm layer of bacterial biomass. (Data was obtained from the laboratory of the presented authors).

**Table 1 sensors-20-04326-t001:** Critical characteristics of various sensor types.

Sensor Type.	Relevant Interference ^a^	LOD, μM	Resp. 90%	Shelf Lifetime	Cont. Operat. ^b^	Sea-Water	Response Scaling	Refs.
**NO_3_^−^ Potentiometric ^c^**	Cl^−^	0.3	4 s	months	months	No	No	[4,5]
**NO_3_^−^, chemical amperometric**	O_2_	0.1	NA	months	≤ hours	No	No	[20]
**NO_3_^−^ Enzym.**	O_2_	0.5		30 days	≤ hours	No	No	[22,23,24,25,26,27]
**NO_3_^−^ bacteria**	N_2_O, NO_2_^−^	0.1	30	30 days	2 weeks	Yes	Yes	[41,42,43,44,45,46,47,49,50,51]
**NO_2_^−^ Potentiometric**	Cl^−^, SCN^−^	1	10 s	?	?	No ^d^	No	[4,5]
**NO_2_^−^, chemical reductive**	O_2_, Br^−^	5	?	3 weeks	≤ hours	No	No	[30]
**NO_2_^−^, chemical oxidative**	None?	0.1	2 s	100 days	≤ hours	No	No	[21]
**NO_2_^−^, enzymatic reductive**	O_2_	0.07	?	30 days	≤ hours	No	No	[40]
**NO_2_^−^, enzymatic oxidative**	I^−^, Br^−^	0.01	2 s	30 days	?	No	No	[38]
**NO_2_^−^, bacteria**	N_2_O	0.1	30	30 days	1 week	Yes	Yes	[42,46,49]
**SO_4_^2−^, Potentiometric**	Cl^−^	0.6	15	?	?	No	No	[52]
**SO_4_^2−^, bacteria**	H_2_S	5	300	?	?	Yes	Yes	This study

^a^ Intereference tests are for most sensors only done for inorganic species. ^b^ Continuous operation for several hours in complex media has only been demonstrated for ionophore- and whole-cell sensors. Times given are best estimates from presented data. ^c^ Several companies, for example Xylem (www.xylemanalytics.com) sell NO_3_^−^ sensors for wastewater etc. that have operational lifetimes of more than a year; microsensors have much shorter lifetimes. The commercial suppliers do not claim 2 s response time and 0.2 µM LOD. Xylem thus claims operational ranges from 7 µM to 30 mM. ^d^ Nitrite sensors have LOD in seawater of 10 µM [4] which is far above natural concentrations.

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
