# Peer review of "Ion Selective Amperometric Biosensors for Environmental Analysis of Nitrate, Nitrite and Sulfate"

_sensors, 2020, doi:10.3390/s20154326_

Round 1
Reviewer 1 Report
The review entitled “Ion selective amperometric biosensors for inorganic ions” from Revsbech et al. as reported the recent advances in amperometric biosensors for the detection of ions such as NO3- , NO2- , NH4+ and SO4 2- by using enzyme or bacteria.
For sure the topic is interesting, especially because the authors investigated aspects that are fundamental for a real application in field, such as the long-term stability, or the application to saline waters.
The language and style are almost fine even if a little revision of the English should be performed as well. Some parts of the paper should be rephrased to be understandable.
I think this perspective can then be published in Sensors after the following minor revisions:
- Have Figure 3 and 4 been previously published? If yes, please, add the references. For Figure 3, please, take care of the labels on the axis…
- The authors should explain better pag 9, where they presented the ESC system (lines 244-255) because these lines are not really clear.
- I suggest to prepare a table that summarize the most relevant sensors reported in the paper, along with their relevant figure of merit (such as the LOD, or the relevant features with respect to environmental monitoring)
- I noticed that the authors focussed on the long-term stability of the sensors, but they mostly avoid to comment the response time of the sensors they presented. Please, try to figure out most of the advantages and drawbacks of the different presented sensors .
Author Response
The review entitled “Ion selective amperometric biosensors for inorganic ions” from Revsbech et al. as reported the recent advances in amperometric biosensors for the detection of ions such as NO3- , NO2- , NH4+ and SO4 2- by using enzyme or bacteria.
For sure the topic is interesting, especially because the authors investigated aspects that are fundamental for a real application in field, such as the long-term stability, or the application to saline waters.
The language and style are almost fine even if a little revision of the English should be performed as well. Some parts of the paper should be rephrased to be understandable.
I think this perspective can then be published in Sensors after the following minor revisions:
- Have Figure 3 and 4 been previously published? If yes, please, add the references. For Figure 3, please, take care of the labels on the axis…
- None of the figures have been published previously. Also reviewer no. 3 mention something about “Chinese font” in Figure 3. The only possibility we see is a transformation problem in the editorial office, so we have saved the picture as a PNG file and inserted it again.
- The authors should explain better pag 9, where they presented the ESC system (lines 244-255) because these lines are not really clear.
We have added a few lines and terms to the section, so that it should be more evident how ESC works.
- I suggest to prepare a table that summarize the most relevant sensors reported in the paper, along with their relevant figure of merit (such as the LOD, or the relevant features with respect to environmental monitoring)
A table is now inserted
- I noticed that the authors focussed on the long-term stability of the sensors, but they mostly avoid to comment the response time of the sensors they presented. Please, try to figure out most of the advantages and drawbacks of the different presented sensors .
Response time is now included in the table . The effect of design and temperature on response time and dynamic range of the bacteria-based biosensors is now treated in the paper
Reviewer 2 Report
To the best of my knowledge, this review is among the first (if not the first one) devoted to the topics of ion selective amperometric biosensors. In fact, the review is focused on four analyte/ions (NO3-, NO2-, NH4+ and SO42-), able to be involved in biological redox processes. Relevant examples based on very recent information were provided.
However, two aspects should be considered:
(i) The title can be better connected to the review content. Indeed, NO3-, NO2-, NH4+ and SO42- are inorganic ions, but their ability to participate in redox biological processes is far more relevant for the review subject.
(ii) The review is structured considering the nature of the detected ion. I believe that a criterion based on the type of the recognition and/or the detection process, involved in the biosensor functioning, could increase the review utility.
Author Response
To the best of my knowledge, this review is among the first (if not the first one) devoted to the topics of ion selective amperometric biosensors. In fact, the review is focused on four analyte/ions (NO3-, NO2-, NH4+ and SO42-), able to be involved in biological redox processes. Relevant examples based on very recent information were provided.
However, two aspects should be considered:
- The title can be better connected to the review content. Indeed, NO3-, NO2-, NH4+ and SO42- are inorganic ions, but their ability to participate in redox biological processes is far more relevant for the review subject.
The title has been changed
- The review is structured considering the nature of the detected ion. I believe that a criterion based on the type of the recognition and/or the detection process, involved in the biosensor functioning, could increase the review utility.
For the chemical and enzymatic sensors it might be possible to make a structure as suggested by the reviewer, but we find the similarities in approaches for nitrite and nitrate too weak to make the change. For the bacteria-based sensors that structure is already there.
Reviewer 3 Report
The review “ Ion selective amperometric biosensors for inorganic ions” summarizes the recent developments in potentiometric ion-selective electrodes applied for detection of ions having important physiological meaning, i.e. NO3-, NO2-, NH4+ and SO42-. However, the authors did not specify in what objects. The entire manuscript is written in a vague and non-adequate manner. The chosen images are not informative and do not help to get closer insights into the topic or challenges in the reported field. The provided references list and examples related to amperometric biosensors development towards NO3-, NO2-, NH4+ and SO42- detection in blood, wastewater and food samples are incomplete.
My major concern within this submission – no attempts to critically analyze the current challenges in a chosen research filed or to propose further perspectives in electrochemical biosensors development employed for NO3-, NO2-, NH4+ and SO42- detection were done. Therefore, I cannot support publication of this review in its present form.
Below the authors will find a list of suggestions for improvements. Probably, following them and trying to address them, the authors can improve the review on focus and clarity.
- Title – should be modified. Something is missing in a title – “…for inorganic ions detection”? From the present title it is absolutely unclear in what objects ion selective biosensor will be applied for detection of inorganic ions. Which ions?
- The authors should decide on which type of sensors they would like to be focused in the present review. Thus, in a title they refer the reader to “amperometric biosensors”, in introduction – to “ion selective electrodes” and in conclusion – to “Ionophore-based potentiometric electrodes”.
- Abstract. Suggest to specify the type of biosensors in the abstract to make the topic clear for the audience. “Inorganic ions that can be redox-transformed by living cells can be sensed by amperometric? Potentiometric? …biosensors..”
- „biologically important ions“ is incorrect and should be replaced by “…for detection of ions having a biological meaning”.
- Introduction The introduction is written in a non-adequate manner. Thus, it is unclear which conventional analytical techniques are used for detection of targeted ions in aqueous solutions, blood or environmental samples, which challenges they have and why potentiometric ion-selective electrodes must be better. The authors mentioned “one option to create very selective sensors for ions…” – if the goal is only to achieve excellent dynamic range, LOD, LLOQ and etc then ICP-MS (ppt level!!!) or IEC-HPLC must be methods of choice instead of biosensors.
- “Concentrations of inorganic ions can be determined by electrochemical techniques” – which ions? Not all ionic species can be determined by electrochemical techniques. In what objects? Please specify.
- Line 36 “Advances in ion selective electrode technology have been reviewed recently [5, 7].” What is the goal of the present review then? Please highlight it.
- Line 50 “…focus will here be on NO3-, NO2-, NH4+ and SO42-“. Why? Can the authors give any specific reason for this choice?
- Parts 2-3 (should be combined in one part). The entire parts are written in “copy-paste” format. Instead, to increase the scientific impact of this review, the authors could try to summarize the achievements towards nitrate detection in more condensed, focused and critical form, viz. group (1) - enzymatic biosensors (nitrite reductase based, horseradish peroxidase, etc) – examples of LOD, LLOQ; their advantages/limitations, impact of matrix effects; (2) - non-enzymatic (nanomaterial-based) – examples, advantages/limitations; (3) hybrid biosensors (hemoglobin and AuNPs, etc) - working principle, advantages/limitations. (4) – Bacteria-based biosensors – working principle, examples, advantages/limitations; (5) macro-scaled biosensors…
The authors could try to summarize all these data in a single table to guide the reader.
- Line 227 – Part “4. Scaling of sensor response by ESC” – it is unclear how the reported technique is related to the developments and challenges in nitrate detection by biosensors?
- Importantly, the sensitive layer immobilizations techniques are missing in this review.
- Line 265, Part “5. Biosensor for sulfate” – only one?
- Moreover, at the beginning the authors have attempted to highlight the advantages of NO3-, NO2- and NH4+ biosensors. However, no examples for NH4+ detection by means of electrochemical biosensors were given. Instead, the authors have decided to stress the operating principle of N2O sensors. Surprisingly, the small part (6-7 sentences only and two (!!) literature references) related to NH4+ sensors appeared at the end of this review, viz. in part 6. “Biosensor for ammonium “.
- Figure 3 – please remove Chinese font.
Author Response
The review “ Ion selective amperometric biosensors for inorganic ions” summarizes the recent developments in potentiometric ion-selective electrodes applied for detection of ions having important physiological meaning, i.e. NO3-, NO2-, NH4+ and SO42-. However, the authors did not specify in what objects. The entire manuscript is written in a vague and non-adequate manner. The chosen images are not informative and do not help to get closer insights into the topic or challenges in the reported field. The provided references list and examples related to amperometric biosensors development towards NO3-, NO2-, NH4+ and SO42- detection in blood, wastewater and food samples are incomplete.
We have changed the title so that it directly refers to environmental analysis of nitrate, nitrite and sulfate. The discussion of ammonium has been taken out, as no applicable biosensors for ammonium have been developed. And yes, we did not mention all the environmental analyses with these sensors, as we have focused on the techniques and not on the many concrete applications. I have given relevant examples showing the potential, but there are many more from our own lab with analysis in biofilms, soils and sediments. I do not understand that the illustrations are not considered informative – they illustrate the sensor principles and examples of use. All illustrations are new and there are therefore no references to former publications.
My major concern within this submission – no attempts to critically analyze the current challenges in a chosen research filed or to propose further perspectives in electrochemical biosensors development employed for NO3-, NO2-, NH4+ and SO42- detection were done. Therefore, I cannot support publication of this review in its present form.
We totally disagree with the reviewer. We have concluded that the amperometric purely chemical sensors and enzymatic sensors do not have sufficient stability to be used for environmental applications – as evidenced by lack of practical use and commercial suppliers. For the bacteria-based sensors we can show extensive environmental use, but no longer a commercial source due to poor lifetime – and we point to the factors that have to be improved to obtain such a long lifetime.
Below the authors will find a list of suggestions for improvements. Probably, following them and trying to address them, the authors can improve the review on focus and clarity.
- Title – should be modified. Something is missing in a title – “…for inorganic ions detection”? From the present title it is absolutely unclear in what objects ion selective biosensor will be applied for detection of inorganic ions. Which ions? Title has been changed so that it is more focused – stressing that we are focusing on environmental analysis. Also ammonium has been taken out – reason given below.
- The authors should decide on which type of sensors they would like to be focused in the present review. Thus, in a title they refer the reader to “amperometric biosensors”, in introduction – to “ion selective electrodes” and in conclusion – to “Ionophore-based potentiometric electrodes”.
The focus is clearly on the biosensors, and within this theme with focus on the bacteria-based sensors, as they are the only ones that have proven usable for detailed environmental investigations. With the change in title and an addition to the abstract this should now be clear.
- Abstract. Suggest to specify the type of biosensors in the abstract to make the topic clear for the audience. “Inorganic ions that can be redox-transformed by living cells can be sensed by amperometric? Potentiometric? …biosensors..”
- Has been changed
- „biologically important ions“ is incorrect and should be replaced by “…for detection of ions having a biological meaning”.
- Has been replaced by “environmentally important ions”
- Introduction The introduction is written in a non-adequate manner. Thus, it is unclear which conventional analytical techniques are used for detection of targeted ions in aqueous solutions, blood or environmental samples, which challenges they have and why potentiometric ion-selective electrodes must be better. The authors mentioned “one option to create very selective sensors for ions…” – if the goal is only to achieve excellent dynamic range, LOD, LLOQ and etc then ICP-MS (ppt level!!!) or IEC-HPLC must be methods of choice instead of biosensors.
Potentiometric ion-selective sensors, also those for nitrate and nitrite, have been used quite extensively for detailed environmental analysis (ref. 4 by de Beer gives an excellent review ) as have the bacteria-based sensors. The alternative nitrite and nitrate sensors have not! It is thus fair to highlight the potentiometric ion-selective sensors, fopr which there are multiple commercial suppliers. It is not our intention to give a review of all available non-sensor based techniques – others have done that. Reference 19 and 20 thus go through all available techniques for nitrite and nitrate analysis.
- “Concentrations of inorganic ions can be determined by electrochemical techniques” – which ions? Not all ionic species can be determined by electrochemical techniques. In what objects? Please specify.
We have corrected the sentence to “Concentrations of many inorganic ions in the environment” to stress that we are focusing on environmental analysis. And yes, sensors have not been developed for all species.
- Line 36 “Advances in ion selective electrode technology have been reviewed recently [5, 7].” What is the goal of the present review then? Please highlight it.
- The word “potentiometric” has now been added – so that it can be distinguished from the current review.
- Line 50 “…focus will here be on NO3-, NO2-, NH4+ and SO42-“. Why? Can the authors give any specific reason for this choice?
We have now taken ammonium out – as there is no practically usable and/or commercially available biosensor for ammonium. I am fully aware that the same argument could be used for enzymatic nitrite and nitate biosensors, but so much work has been done on those that it is fair to give an evaluation of their performance. Nitrite, nitrate and sulfate are the most important (after oxygen) oxidants in nature, and their transformations in nature are studied intensively. A search on WOS on the words “denitrification” and “sulphate reduction” both resulted in more than 30.000 hits – illustrating how important quantification of these species is. I guess that our laboratory alone has published about 30 papers using nitrate and nitrite biosensors, resulting in important new insight into the functioning of nature. We have only given a few illustrative examples in the review, but we could have mentioned as example the finding of denitrification in eucaryotes (Nature 443:93-96). We do not feel that the review (or rather “perspective”) should focus on applications as such, but on potential for application!
- I am aware that the sulfate sensor is very preliminary, and only illustrates the potential. However, these are completely new data, and we decided to include it here. The Corona crisis stopped furher development of this sensor for a long period, but after the summer we will continue. I have no doubt that we will obtain a very useful sensor!
- Parts 2-3(should be combined in one part). The entire parts are written in “copy-paste” format. Instead, to increase the scientific impact of this review, the authors could try to summarize the achievements towards nitrate detection in more condensed, focused and critical form, viz. group (1) - enzymatic biosensors (nitrite reductase based, horseradish peroxidase, etc) – examples of LOD, LLOQ; their advantages/limitations, impact of matrix effects; (2) - non-enzymatic (nanomaterial-based) – examples, advantages/limitations; (3) hybrid biosensors (hemoglobin and AuNPs, etc) - working principle, advantages/limitations. (4) – Bacteria-based biosensors – working principle, examples, advantages/limitations; (5) macro-scaled biosensors…
The authors could try to summarize all these data in a single table to guide the reader.
We have now inserted a table summarizing the most important characteristics of the various sensor classes. We will not go into the very detail of the various enzymatic biosensors, as none of them (in contrast to ionophore-based and bacteria-based sensors) have proven useful for environmental analysis. It has been shown that that they can quantify nitrate/nitrite at relatively high concentrations in substances like vegetable juice, sausage and milk, but only by short term experiments, and there are no commercial suppliers or studies where they are applied for more detailed environmental analysis or monitoring. It is actually quite difficult to give a fair comparison in a table, as for example detection limits usually are obtained in pure buffer solution, and ability for continuous use is never mentioned for enzymatic or purely chemical sensors.
- Line 227 – Part “4. Scaling of sensor response by ESC” – it is unclear how the reported technique is related to the developments and challenges in nitrate detection by biosensors?
- Scaling of the response is very important for environmental analysis. By this technique it is possible to use the same sensor for both low and high concentration ranges, and even to do an in situ baseline calibration. This now stressed in the text. The technique has been used in many environmental studies – for example in reference 49.
- Importantly, the sensitive layer immobilizations techniques are missing in this review.
- For the bacteria-based biosensors we have mentioned that immobilization attempts (agar, alginate) have been unsuccessful. For the enzyme-based sensors there are many modification, and it is not the intention of this review to describe these details – but to summarize what came out of it. And our conclusion is that none of these techniques are environmentally useful as they do not have sufficient stability/lack of interferences.
- Line 265, Part “5. Biosensor for sulfate” – only one?
I have, of course, searched on WOS for sulfate biosensors, and could not find any. Sulfate is very difficult to activate, so purely enzymatic sensors for sulfate may not be possible. So I guess that the data for a sulfate response of a biosensor presented in Figure 7 are completely novel.
- Moreover, at the beginning the authors have attempted to highlight the advantages of NO3-, NO2- and NH4+ biosensors. However, no examples for NH4+ detection by means of electrochemical biosensors were given. Instead, the authors have decided to stress the operating principle of N2O sensors. Surprisingly, the small part (6-7 sentences only and two (!!) literature references) related to NH4+ sensors appeared at the end of this review, viz. in part 6. “Biosensor for ammonium “.
- We agree with the reviewer – there is no need to mention a biosensor that cannot be used in the environment. So it is taken out.
- Figure 3 – please remove Chinese font.
- We do not see any Chinese font in Figure 3 – so I do not understand the request for correction. The only possibility we see is a transformation problem in the editorial office, so we have saved the picture as a PNG file and inserted it again.
Round 2
Reviewer 3 Report
The authors have partly improved their manuscript on focus and clarity. However, they mentioned in the text that “some contamination and immobilization problems” occur, they did not specify which immobilization protocols can be used and which problems related to “immobilization” can be expected in order to support the discussion in a manner typical for the Perspectives. Thus, depending on the used immobilization protocol, different leakage of enzyme can be observed resulting in the different rates of fouling by organisms growing. Moreover, namely the utilized immobilization protocol often determines the life time of amperometric biosensors. This fact cannot be simply ignored.
In conclusion it is mentioned „It should be possible to extend the lifetime of such sensors to several months”. However, the authors did not specify how this ambitious goal can be achieved. If the authors propose to achieve it by means of scaling of sensor response by ESC, they should stress it in the conclusion.
In addition, if all image belong to the authors, they should specify it, i.e. by the sentence “Original image” or “Data obtained in the laboratory of the presented authors”. Also, the experimental data related to these images should be provided, see Figure 3,4,6,7.
Author Response
Please see the detailed response in the PDF file attached.
